# Dynamics of the Gut Microbiome and Transcriptome in Korea Native Ricefish (*Oryzias latipes*) during Chronic Antibiotic Exposure

**DOI:** 10.3390/genes13071243

**Published:** 2022-07-14

**Authors:** Ju Bin Yoon, Sungmin Hwang, Jun Hyeok Yang, Seungki Lee, Woo Young Bang, Ki Hwan Moon

**Affiliations:** 1Division of Convergence on Marine Science, Korea Maritime & Ocean University, Busan 49112, Korea; fishbin12345@gmail.com (J.B.Y.); ardim4100@gmail.com (J.H.Y.); 2Ocean Science and Technology School, Korea Maritime & Ocean University, Busan 49112, Korea; 3Clean Energy Research Center, Korea Institute of Science and Technology, Seoul 02792, Korea; sungminhwang@kist.re.kr; 4Department of Marine Bioscience and Environment, Korea Maritime & Ocean University, Busan 49112, Korea; 5National Institute of Biological Resources, Environmental Research Complex, Incheon 22689, Korea; metany@korea.kr

**Keywords:** ricefish, microbiome, ampicillin, erythromycin, immune and stress-related genes

## Abstract

Antibiotics have been widely used to inhibit microbial growth and to control bacterial infection; however, they can trigger an imbalance in the gut flora of the host and dysregulate the host gene regulatory system when discharged into the aquatic environment. We investigated the effects of chronic exposure to a low concentration of erythromycin and ampicillin, focusing on gut microbiome and global gene expression profiles from Korea native ricefish (*Oryzias latipes*). The proportion of *Proteobacteria* (especially the opportunistic pathogen *Aeromonas veronii*) was significantly increased in the ricefish under the chronic exposure to erythromycin and ampicillin, whereas that of other bacterial phyla (i.e., *Fusobacteria*) decreased. In addition, the expression of genes involved in immune responses such as chemokines and immunocyte chemotaxis was significantly influenced in ricefish in the aquatic environment with antibiotics present. These results show that the internal microbial flora and the host gene expression are susceptible even at a low concentration of chronic antibiotics in the environment, supporting the importance of the appropriate use of antibiotic dose to maintain the sustainable and healthy aquaculture industry and water ecosystem.

## 1. Introduction

Antibiotics are a substance used to prevent or treat diseases caused by bacteria by inhibiting the growth of bacteria with several action mechanisms, such as impeding cell wall biosynthesis and translation machinery. For this reason, various antibiotics are widely used in livestock farms, hospitals, aquaculture, veterinary hospitals, agriculture, household products, research, and industrial fields [1]. As the mass use of antibiotics grows, antibiotics are being detected in wastewater treatment plants, freshwater and seawater, biological solids, sediments, soil, and aquatic organisms [2]. Even feces from animals that have been injected with antibiotics are excreted into the aquatic environment through drainage systems and cause damage to aquatic environments [3,4]. Antibiotics released into the environment are diluted and affect bacteria at low concentrations, which would not be fatal to the bacteria; however, this acts as selective pressure that can lead to negative consequences such as the acquisition of antibiotic-resistant genes and uncontrollable regulation of virulence genes [5,6,7]. In addition, antibiotics act on the host as well, affecting changes in behavior, growth, development, reproduction, mortality, and essential flora [8,9,10].

The gut microbiome is a community of bacteria in the gut and is known to be interactive with the host, resulting in various roles such as developing the host immune system, providing useful metabolites, strengthening the barrier of the intestine, and regulating gene expression [11]. This gut microbiome is distinctive among species and changeable by various external stimuli as well as biological factors such as diet, hunger, and age [12,13,14,15,16,17]. It is known that dysbiosis, in which the microbial diversity is rapidly reduced by certain stress conditions in the intestine and the expansion of certain bacteria is promoted, can cause metabolic disorders such as obesity and inflammatory bowel disease [18].

Ricefish (*Oryzias latipes*) are one of the best model organisms of fish, along with zebrafish (*Danio rerio*), due to the completion of their whole genome sequence and the relatively short period (2–3 months) required for them to reach maturity due to the absence of a spawning period [19]. The Korean native ricefish used in this study were caught in nature without undergoing breeding, and this is a promising viable tool to study the wild type ricefish for presenting the effects of antibiotics on the host in freshwater.

We sought to understand the effect of residual antibiotics on the aquatic environment and, in particular, on fish in freshwater. To verify this, ampicillin (β-lactam antibiotics) and erythromycin (macrolide antibiotics) were selected, since these two representative antibiotics are widely used in various purposes and detected in environmental specimens [2,20], and they were supplemented at relatively low concentrations to an experimental aquarium. The Korea native ricefish that were exposed to the antibiotics had their gut microbial flora examined and the alteration of their gene expression monitored. Overall, this study will provide both the potential risk and the fundamental interactions between the gut microbiome and host transcriptome at persistent low-dose antibiotics.

## 2. Materials and Methods

### 2.1. Experimental Animals

Korean native ricefish (*O. latipes*) were supplied from the National Institute of Biological Resources (NIBR) (Incheon, Republic of Korea). Fish were caught from the Naerincheon, Hongcheon, Gangwon-do in the Republic of Korea (37.9536, 128.3121) to determine how antibiotics influence native living organisms and aquatic environments, and were stabilized in the NIBR laboratory for 4 weeks. The average weight (±SD) and average length (±SD) of the ricefish used in this study were 0.27 ± 0.05 g and 2.88 ± 0.12 cm, respectively. The surviving individuals were procured for subsequent experiments including microbiome and transcriptome in our laboratory at the Korea Maritime & Ocean University. The procured fish from NIBR were acclimatized for 2 weeks in approximately 15 L (25 cm long, 25 cm wide, 25 cm deep) water tanks with aerated dechlorinated tap water and a natural photoperiod of 14 h light and 10 h dark before the experiment. The water parameters were monitored daily to maintain the following conditions: water temperature (27 ± 2 °C), pH (7.5 ± 0.3), and dissolved oxygen (7.6 ± 0.3 mg/L). During the acclimatization, fish were fed with a commercial diet (PRODAC, Cittadella, PD, Italy) once per day.

### 2.2. Chronic Antibiotics Exposure Assay

Four liters of aerated dechlorinated tap water was added to three glass aquariums (approximately 5 L volume at 15 cm long, 22 cm wide, and 17 cm deep). Using antibiotics stock, three water conditions were prepared: non-treated, Ery (3.9 μg/mL), and Amp (3.125 μg/mL). Ten ricefish were exposed to antibiotics in each of the three conditions (control, Ery, Amp) during 30 days under semi-static conditions. The tank water was refreshed twice a week by discarding 2 L of the old breeding water and adding 2 L of the new breeding water with the same antibiotic concentration as the old water. Moreover, every two weeks, all breeding water was refreshed. During the exposure, fish were fed with a commercial diet (PRODAC, Cittadella, PD, Italy) once per day.

### 2.3. Gut Microbe Isolation and 16S rDNA Sequencing

Three ricefish were randomly selected and anesthetized using tricaine methanesulfonate (Sigma-Aldrich, Burlington, MA, USA). Fish were washed with 70% ethanol to remove surface bacteria and washed once more with 1X phosphate buffered saline (PBS), then their guts were dissected and homogenized with 10 mL of 1X PBS. After diluting tenfold with 1X PBS, the homogenates were spread by 100 μL of nutrient agar (NA), Luria-Bertani (LB) agar, tryptic soy agar (TSA), brain heart infusion (BHI) agar, and Lactobacilli MRS (MRS) agar, respectively. Separate media were cultured at 27 °C for a maximum of 2 days. Twenty-five strains of bacteria were isolated from separate media, then finally six strains were isolated by comparing the colony phenotypes of the isolates. Six gDNA isolates were extracted using a HiGene™ Genomic DNA Prep Kit (BIOFACT, Daejeon, Korea) according to the manufacturer’s instructions and were quantified using an Epoch 2 Microplate Spectrophotometer (BioTek, Winooski, VT, USA). 16S rDNA sequencing was conducted at DNALINK, Inc. (Seoul, Korea). The six isolates were stored at −80 °C.

### 2.4. Minimum Inhibitory Concentration (MIC) Test against O. latipes Gut Isolates

To set up the antibiotic concentration for the chronic antibiotic exposure assay, we conducted a MIC test against six *O. latipes* gut isolates. Before the experiment, all isolate strains were streaked on TSA and incubated at 27 °C for 18 h. Then, cultured single colonies were pre-cultured on 2 mL tryptic soy broth (TSB) at 27 °C for 18 h. Stock solutions were prepared by dissolving erythromycin (Ery) (Sigma Aldrich, Burlington, MA, USA) in absolute ethyl alcohol and ampicillin (Amp) (Generay Biotech, Shanghai, China) in ddH_2_O. To perform the MIC test, antibiotic-treated TSB was prepared by using two-fold serial dilutions from maximum concentrations of Ery (62.5 µg/mL) and Amp (100 µg/mL). All the diluted medium was aliquoted into a 96-well plate in 150 µL, and the pre-cultured strains were inoculated by 1% (*v*/*v*) each. The 96-well plate was incubated at 27 °C within the Epoch 2 Microplate Spectrophotometer (BioTek, Winooski, VT, USA) for 18 h and optical density (OD) at 600 nm was used to measure cell growth every 20 min.

### 2.5. O. latipes Gut Microbiome Analysis

Two or three ricefish exposed to chronic antibiotics were randomly selected from each group and anesthetized using tricaine methanesulfonate. Bacterial genomic DNA was extracted from the fish guts dissected by using the HiGene™ Genomic DNA Prep Kit (BIOFACT, Daejeon, Korea) according to the manufacturer’s instructions. Specific primer pairs (Bakt_341F 5′-CCTACGGGNGGCWGCAG-3′ and Bakt_805R 5′-GACTACHVGGGTATCTAATCC-3′) that recognize the hypervariable V3–V4 region were used to prepare for a library construction. High-throughput sequencing was performed using the MiSeq platform (Illumina, San Diego, CA, USA), followed by a quality check including the removal of the adapter and low sequence scores (<20) via FastQC v.0.11.7 and Cutadapt v.1.18. The preprocessed sequence data were clustered by using CD-HIT-OTU with a cutoff value at 0.03, which is equivalent to an over 97% sequence similarity score for operational taxonomic units (OTUs) [21], followed by further analysis for the α-diversity including Shannon and Simpson (relative dominance of some species) indices in each sample, as well as for the rarefaction curves and PCoA (principal coordinate analysis) plots in OTU-based groups under the QIIME2 platform [22]. SILVA reference database version 138 was used to classify the bacterial taxonomy from the phylum to species levels [23].

### 2.6. Differentially Expressed Genes (DEGs) Analysis

Three ricefish were randomly selected from each group, anesthetized using tricaine methanesulfonate, and homogenized to extract total RNA with TRIzol™ Reagent. The purified total RNA was further processed to construct an mRNA sequencing library according to the manufacturer’s instructions (Illumina TruSeq stranded mRNA library prep kit, San Diego, CA, USA). mRNA was purified and fragmented from total RNA (1 μg) using poly-T oligo-attached magnetic beads, then reverse transcribed into cDNA with adapters. The Illumina Novaseq 6000 sequencing system was used to perform high-throughput sequencing that generated average 70 million reads with 2 × 100 bp read length in each sample. TopHat v.2.0.13 [24] was used to map reads to the reference genome of *O. latipes*, followed by the identification of differentially expressed genes (DEGs) with the default options of Cuffdiff v.2.2.0 [25]. For ontology analysis, DEGs (log2 fold change larger than 1 and a false discovery rate less than 0.05) were applied to DAVID as an input to obtain a comprehensive set of functional annotation [26].

### 2.7. Statistical Analysis

Statistical analyses used in this study were performed by Prism 9.0 (GraphPad, San Diego, CA, USA). The statistical significance was determined by *t*-test. The significance level was defined as *p* values of ≤ 0.05.

## 3. Results

### 3.1. Determination of Exposure Concentration of Ampicillin and Erythromycin in O. latipes Gut Isolates

Antibiotics are widely used to prevent the growth of microorganisms. The inhibitory molecules that are released into the environment are diluted and present at low concentrations. This may not be fatal to bacteria; however, it may act as a selective pressure that can lead to negative consequences such as the acquisition of antibiotic-resistant genes and uncontrollable regulation of virulence genes. Considering this, we firstly hypothesized that the gut microbes of ricefish would have different antibiotic susceptibility depending on the species. Therefore, we tried to determine the exposure concentration of antibiotics based on the antibiotic susceptibility of representative bacteria isolated from the gut of ricefish. We confirmed that six gut microorganisms separated from the intestines of Korean native ricefish were *Shewanella xiamenensis*, *Flavobacterium* sp., *Microbacterium* sp., *Aeromonas hydrophila*, *Aeromonas* sp., and *Bacterium* strain BS2147, respectively, through 16S rDNA sequencing. In addition, the minimum inhibitory concentration (MIC) test for ampicillin and erythromycin was performed in the six identified strains, respectively. Based on the results of the MIC test and the concentration of antibiotics used in the aquaculture (data not shown) [27], we determined the final exposure concentration of antibiotics as 3.9 μg/mL for Ery and 3.125 μg/mL for Amp.

### 3.2. Effects of Chronic Antibiotic Exposure on the Richness and Diversity of the Gut Microbiome

We evaluated α-diversity to determine how the richness and diversity of the gut microbiome of ricefish was altered by chronic exposure to antibiotics. When comparing with the control group in Figure 1, we confirmed the decrease of average operational taxonomic units (OTUs) in both the ampicillin and erythromycin treatment groups. The rarefaction curve created on the species observed in Figure 2 gradually reached a plateau in all groups, indicating that the amount of sequence data to identify all species included in the sample was adequate. In addition, species richness decreased in the ampicillin and erythromycin treatment groups because the curve was less steep than that of the control group. As shown in Table 1, the Good’s Coverage index of all the groups was observed to be larger than 99.9%, indicating the sequencing result is reliable. The inverse Simpson index was confirmed to have no significant difference in all the groups (*p* > 0.05). Chao1 was confirmed to significantly decrease in the ampicillin treatment group when compared to the control group (*p* < 0.01). The Shannon index was confirmed to significantly decrease in the erythromycin treatment group when compared to the control group (*p* < 0.05) (Table 1). These results show that the chronic exposure of all the selected antibiotics decreases both the richness and diversity of the intestinal microbial community.

### 3.3. Effect of Chronic Exposure to Antibiotics on the Structure of the Gut Microbial Community

In order to compare the entire structure of the intestinal microbial community, β-diversity was evaluated with PCoA. As shown in Figure 3, each group was confirmed to have a separate microbiome cluster. This indicates the structure of the microbiome was significantly modified by the antibiotic chronic exposure because clusters of the ampicillin and erythromycin treatment groups were remote from the clusters of the control group.

### 3.4. Changes in Relative Abundance of the Gut Microbiome under the Chronic Exposure to Antibiotics

To determine whether dysbiosis is induced by the chronic exposure to antibiotics in the gut of ricefish, we investigated the proportion of altered gut microbiota. As an analysis result of the gut microbiome of ricefish which have been chronically exposed to erythromycin, it was confirmed that the *Proteobacteria* proportion (16.55% increase, *p* < 0.0001) was increased the most at the phylum level compared to that of the control group (Figure 4A). Conversely, the decrease of the remaining phylum including *Fusobacteria* (4.03% decrease, *p* > 0.05) was confirmed, showing the imbalance that the variety of intestinal flora decreases and only a few floras have dominance (Figure 4A). In addition, when compared to the details at the species level, it was confirmed that the proportion of *Aeromonas veronii* (21.29% increase, *p* < 0.0001) increased, the proportion of *Cetobacterium somerae* (4.03% decrease, *p* > 0.05) decreased, and almost all other species decreased (Figure 4B). As a result of the gut microbiome analysis of ricefish chronically exposed to ampicillin, it was found that the proportion of *Proteobacteria* (53.95% increase, *p* < 0.0001) increased by almost two times at the phylum level compared to the control group (Figure 4A). However, the remaining phylum including *Fusobacteria* (45.95% decrease, *p* < 0.0001) showed a decreasing trend, confirming the imbalance of gut flora in ricefish which are chronically exposed to erythromycin (Figure 4A). In addition, the proportions of *Aeromonas veronii* (28.6% increase, *p* < 0.0001) and *Vibrio parahaemolyticus* (25.92% increase, *p* < 0.0001) were increased, respectively, whereas the proportion of *Cetobacterium somerae* (41.92% decrease, *p* < 0.001) was decreased (Figure 4B).

### 3.5. Changes in the Expression of Stress and Immune-Related Genes in Ricefish Chronically Exposed to Antibiotics

To determine whether a specific pathogen becomes a dominant species in the gut of ricefish by chronic exposure to antibiotics and increases infection in ricefish, the transcription levels of host stress and immune-related genes were identified. We investigated any change in the gene expression of ricefish chronically exposed to ampicillin and erythromycin by RNA-seq as well as by gene ontology analysis. The number of genes showing significant expression changes in the three aspects of molecular function (MF), cellular component (GC), and biological process (BP) is shown in Appendix A, and it was confirmed that the expression of various stress and immune-related genes was changed in the ricefish when ampicillin and erythromycin were chronically present (Figure 5). Among them, gene expression for genes responsive to TNF, IL-1, and IFN-γ, for lymphocyte and monocyte chemotaxis (BP), and for chemokine receptor binding (MF) was increased compared to the control group. Furthermore, the number of genes whose expression was changed and the width of expression change in the erythromycin-treated group were larger than those of the ampicillin-treated group. The upregulated genes indicate that chronic exposure to both ampicillin and erythromycin causes the inflammatory response in ricefish. Specifically, in ricefish chronically exposed to erythromycin, it was found that the chemokine-mediated signaling pathway, neutrophil chemotaxis, antigen processing and presentation, response to oxidative stress, inflammatory response, defense response-related gene expression (BP), chemokine activity, peroxidase activity-related gene expression (MF), and the expression of MHC class II protein complex-related genes (CC) were further increased, resulting in a stronger inflammatory response. In ricefish chronically exposed to ampicillin, tissue regeneration and the cytokine-mediated signaling pathway-related gene expression (BP) also increased. On the contrary, it was confirmed that blood coagulation, myeloid dendritic cell differentiation, response to tumor necrosis factor, response to lipopolysaccharide, complement activation-related gene expression (BP), serine-type endopeptidase activity (MF) gene expression, membrane attack complex, and fibrinogen complex (CC) decreased. This indicates that other inflammatory response types are caused by the host depending on the antibiotic type.

## 4. Discussion

The balance of the microbiome in the gut of the exposed host as well as the underwater microbiome is being damaged by the antibiotics emitted to the aquatic environment. Numerous studies have suggested that the intestinal microbiome can be changed quickly and significantly due to environmental changes, and that the changed intestinal microbiome can have various effects on the host [28,29,30,31,32]. However, studies that reveal the correlation between altered gut microbiota and host specific gene expression are lacking. This study investigated the relevant effect by exposing Korean native ricefish chronically to a low concentration of antibiotics. We predicted that the low concentration of antibiotics can cause dysbiosis of the gut microbiome of the host, and affect the expression of the host’s stress and immune-related genes.

We selected ampicillin and erythromycin, antibiotics that are used in various fields such as the livestock industry, aquaculture, and hospitals and are continuously detected in the environment, as representative antibiotics [2,20]. In addition, after selecting and confirming six representative bacteria isolated from the intestines of Korean native ricefish, the exposure concentrations of ampicillin and erythromycin were determined by conducting the MIC test. For the concentration of selected antibiotics, the host is thought to be continuously exposed to the selected antibiotic concentration sufficiently when it is detected at 1 mg/L or more in industrial waste water and some rivers [33,34]. As a result of the α-diversity analysis in ricefish chronically exposed to the selected concentration of antibiotics, it was confirmed that the Chao1 index was reduced significantly in the ampicillin treatment group compared to the control group, while the Shannon index was reduced significantly in the erythromycin treatment group (Table 1). In addition, when comparing with the control group, it was confirmed that average OTUs were reduced in both the ampicillin and erythromycin treatment groups (Figure 1), and the same trend was also observed in the rarefaction curve (Figure 2). As a result of the β-diversity analysis, it was confirmed that the cluster distance was remote in both the ampicillin and erythromycin treatment groups compared to the control group (Figure 3). These results indicate that chronic exposure to antibiotics causes a change to the gut microbiome of ricefish, reducing the diversity, and disrupts the balance of the gut microbial community which used to be maintained.

Putting together the results of the relative abundance analysis of the gut microbiota in ricefish, where *Proteobacteria* and *Fusobacteria* were dominant, *Proteobacteria* became more dominant with chronic exposure to erythromycin and ampicillin, whereas most other phyla including *Fusobacteria* were decreased (Figure 4). It is known that the gut microbiota of a healthy host remains stable over time, allowing for symbiotic interactions with the host, such as maintaining gut homeostasis and developing the immune system, whereas in dysbiosis, in which the *Proteobacteria* dominate, it can lead to intestinal inflammation [35]. Although altered microbial diversity tends to partially recover over time, it is also known that opportunistic pathogens can significantly increase under certain environmental conditions [36]. In addition, *Proteobacteria* increased by chronic exposure to antibiotics is a phylum containing various opportunistic pathogens and pathogens, including *Vibrio parahaemolyticus* and *Aeromonas veronii*, which we have focused on, and it is generally known that there are many cases of antibiotic resistance in the environment [37,38]. Therefore, if *Proteobacteria* survive through resistance and become dominant in the host’s guts after being chronically exposed to antibiotics, they can cause diseases by being able to invade the inside of the host through increased intestinal permeability without barrier action through competition with other bacteria [39]. From this point of view, it was confirmed that the proportion of *Cetobacterium somerae*, a non-pathogenic bacterium which used to be dominant in the essential flora of ricefish chronically exposed to both erythromycin and ampicillin, decreased and, on the contrary, the proportion of *Aeromonas veronii*, an opportunistic infectious bacterium, increased significantly (Figure 4). In addition, in the gut flora of ricefish chronically exposed to ampicillin, the pathogen *Vibrio parahaemolyticus* was also increased (Figure 4). However, *Vibrio parahaemolyticus* was not detected in the gut flora of ricefish chronically exposed to erythromycin (Figure 4). This phenomenon is predicted to occur because *Vibrio parahaemolyticus* has different susceptibility depending on the type of antibiotic or because other microbial communities become more dominant species due to antibiotic exposure. It is known that *Aeromonas veronii* and *Vibrio parahaemolyticus* can kill the host by causing disease in the infected fish [40,41], and they are estimated to cause the disease through infection because of the intestinal permeability of the host when the opportunistic infectious bacterium and the pathogen become the dominant species in the intestine.

To confirm this, gene ontology analysis was performed based on RNA-sequencing in ricefish, which had dysbiosis in the intestinal microbiome due to exposure to ampicillin and erythromycin. As a result, it was confirmed that the expression of various stress and immune-related genes was changed in ricefish exposed to ampicillin and erythromycin, respectively (Figure 5). Genes with increased expression in ricefish chronically exposed to ampicillin and erythromycin, respectively, were identified as cellular responses to TNF, IL-1, and IFN-γ, lymphocyte and monocyte chemotaxis (BP), and chemokine receptor binding (MF) (Figure 5). Cytokine is a major regulator of the immune system and is known to induce inflammatory signals that regulate the ability of macrophages to destroy pathogens that have infected hosts [42]. Among them, IL-1 is known to be involved in the regulation of acute and chronic inflammation by microbial infection [43], and IFN-γ is produced by activated T cells and NK cells, which play a role in immune enhancement and regulation [44]. In addition, TNF is known to play various roles related to inflammation, apoptosis, and stimulation of the immune system [45]. Therefore, the increase in the expression of cellular responses to these cytokines in the fish host by chronic exposure to antibiotics supports our hypothesis that an inflammatory response is induced by infection with specific bacteria in a host with dysbiosis. In addition, it was confirmed that the expression of genes related to the chemotaxis of lymphocytes, which are immune cells, and monocytes, which are phagocytes, and genes related to chemokine receptor binding also increases in common, confirming the overall immune system is activated if dysbiosis occurs due to chronic antibiotic exposure. The expression of other stress and immune-related genes showed different patterns in ricefish exposed to ampicillin and erythromycin (Figure 5), confirming that different types of inflammatory reactions were induced in ricefish depending on the type of antibiotic. Specifically, a significant increase was observed in the expression of more stress and immune-related genes in ricefish exposed to erythromycin compared to ampicillin, and in particular, it was confirmed that the expression of MHC class II complex increased strongly (Figure 5). MHC class II is a cell surface glycoprotein that plays a central role in the immune system through exogenous pathways by presenting peptides to antigen receptors in CD4^+^ T cells [46]. The function of MHC class II in bony fishes and mammals is similar, and the expression of MHC class II is known to be upregulated after immune stimulation [47]. Therefore, an increase in MHC class II expression also implies an increase in infection in the host of extracellular pathogens. Additionally, the expression of genes related to the response to oxidative stress was also increased in ricefish chronically exposed to erythromycin. It is known that the production of reactive oxygen species (ROS) is induced by infection with viruses, bacteria, and parasites [48], and the increase in the expression of the host response to oxidative stress induced by ROS can predict the increase in infection in the host as well.

This study confirmed the results that check the microbiome’s diversity (OTUs, α-diversity, β-diversity) and the relative abundance in the intestine, and the ampicillin treatment was found to have a bigger influence on the intestinal microbiome of the fish than the erythromycin treatment. However, the gene ontology results showed the erythromycin treatment group had a stronger immune response than the ampicillin treatment group. These results are expected because the pattern of dysbiosis in the intestinal microflora is different depending on the type of antibiotic due to chronic exposure. Therefore, we confirmed that the influence on the host can be different depending on which flora becomes the dominant species and the competing species that is being reduced. In addition, we could additionally confirm that the expression of various genes related to metabolism, reproduction, DNA replication, RNA transcription, mitosis, hormones, etc., was also changed in ricefish, where dysbiosis occurred due to chronic exposure to antibiotics (Appendix A), which indicates that imbalance in the gut microbiome has a variety of negative effects on host homeostasis and health.

The ricefish used as hosts in this study were captured from the environment, and if the effects of chronic exposure to antibiotics are reproduced in hosts in the actual environment, it can cause an enormous loss to fishery and aquaculture due to disease induction, and additional side effects are expected to occur from humans consuming fish. Further studies can visually confirm the inflammatory response in the intestines of fish chronically exposed to antibiotics, the change in host sensitivity through pathogen infection in fish with dysbiosis from antibiotics, and verify the expression of immune-related factors from fish chronically exposed to antibiotics; furthermore, it is considered to be possible to establish more diverse pathological effects of intestinal microbiological imbalance on the host caused by chronic exposure to antibiotics.

In this study, we suggested the fact that the chronic exposure to antibiotics at low concentrations remaining in the aqueous environment can not only induce dysbiosis of the intestinal microbiome in the host and cause a sensitivity change of the disease, but can also have negative effects on host homeostasis and entire health. Therefore, low concentrations of antibiotics remaining in the environment have various negative effects not only on bacteria but also on the host, providing scientific evidence for the importance of controlling the release of antibiotics. Furthermore, the interpretation of our findings cannot be definitive because the interactions between the gut microbiota and the host are very complex, but our study provides additional insight into the host–microbiome interactions.

## Figures and Tables

**Figure 1 genes-13-01243-f001:**
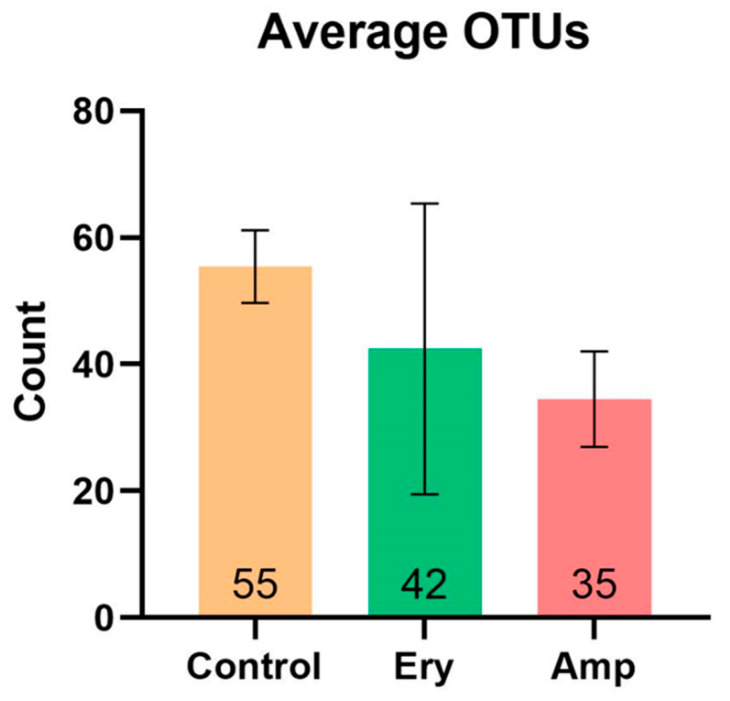
Effects of chronic antibiotics exposure on gut bacterial community shown as average OTUs.

**Figure 2 genes-13-01243-f002:**
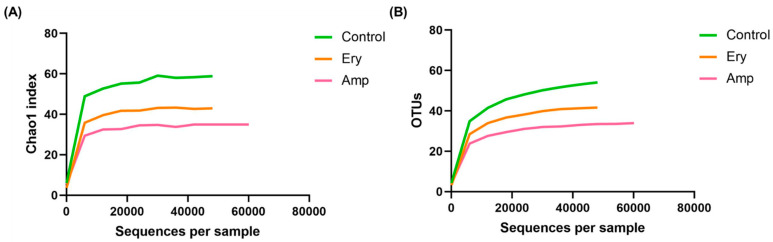
Effect of chronic antibiotics exposure on rarefaction curve: (**A**) chao1 index and (**B**) OTUs.

**Figure 3 genes-13-01243-f003:**
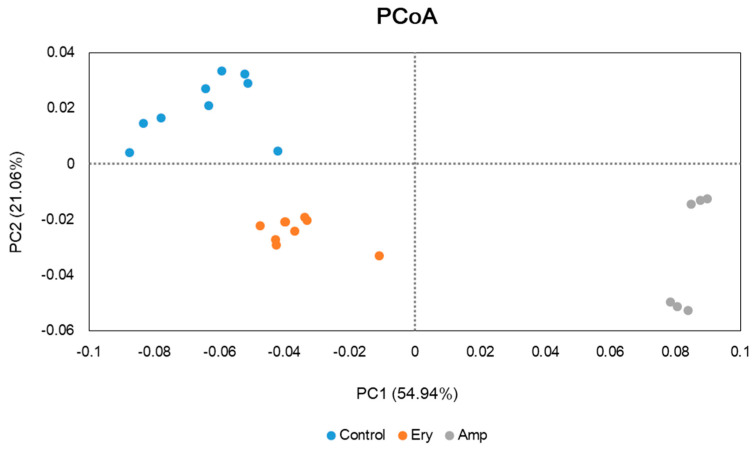
Effect of chronic antibiotics exposure on β-diversity based on the principal coordinate analysis (PCoA) in the gut microbiota. Each point represents a sample with colors representing different groups.

**Figure 4 genes-13-01243-f004:**
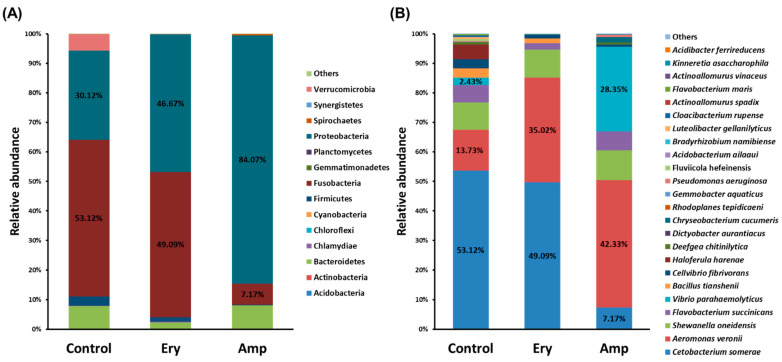
Relative abundance of gut microbiome in ricefish chronically exposed to antibiotics: (**A**) phylum level and (**B**) species level. Each bar represents the average relative abundance of each bacterial taxa for a treatment group.

**Figure 5 genes-13-01243-f005:**
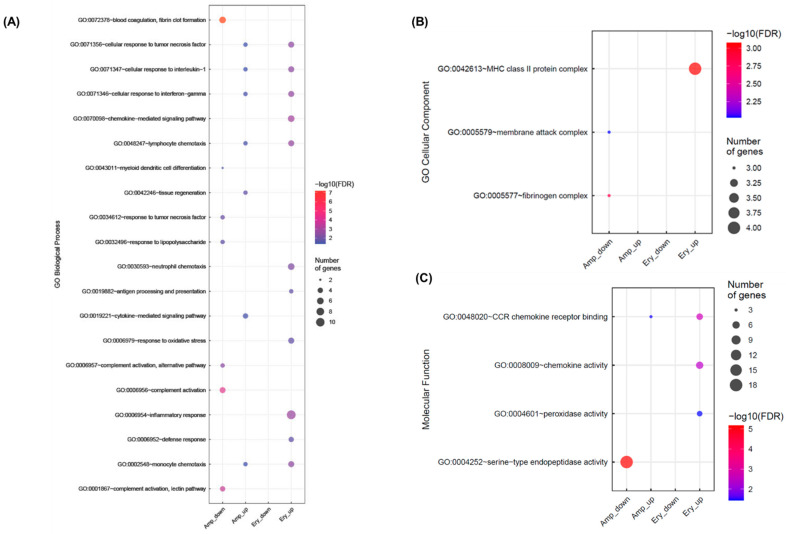
Changes in stress and immune-related gene expression from ricefish by the chronic exposure to antibiotics: (**A**) biological process, (**B**) cellular component, and (**C**) molecular function. The size and color of the dots indicate the number and expression level of DEGs whose expression is changed in each treatment group, respectively.

**Table 1 genes-13-01243-t001:** Effects of chronic exposure to antibiotics on gut bacterial diversity indexes.

Treatment	Control	Ampicillin	*p* Value	Erythromycin	*p* Value
Chao1	60.33 ± 5.83	35.42 ± 7.66	0.001	43.52 ± 22.95	0.051
Shannon	2.42 ± 0.3	2.2 ± 0.03	0.087	1.78 ± 0.1	0.0001
Inverse Simpson	0.67 ± 0.08	0.68 ± 0.01	0.887	0.62 ± 0.02	0.096
Good’s Coverage	0.99 ± 0	0.99 ± 0	0.0001	0.99 ± 0	0.0001

* Note: Values are presented as means ± SD. Differences between each treatment group and control group were analyzed by *t*-tests, significant difference at *p* < 0.05.

## Data Availability

The sequencing raw data discussed in this publication have been deposited in the NCBI through accession number PRJNA852666 and GEO Series accession number GSE205468.

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
