# Peer review of "Dynamics of the Gut Microbiome and Transcriptome in Korea Native Ricefish (Oryzias latipes) during Chronic Antibiotic Exposure"

_genes, 2022, doi:10.3390/genes13071243_

Round 1

Author Response

Response to the Reviewers’ comments

Re: Manuscript genes-1813719 (Dynamics of Gut Microbiome and Transcriptome in Korea Native Ricefish (Oryzias latipes) during Chronic Antibiotics Exposure).

We would like to thank the Reviewers for careful reading of our manuscript and providing helpful comments. We address each of these comments below and feel that the revised manuscript has been improved.    

Reviewers’ comments are copied below (in black), followed by the authors’ point by point responses (in blue). Changes refer to lines in the revised manuscript.

Please see the attachment file.

Reviewer 2 Report

1. Abstracts are not ok for me. The research background is too much, and the research results are not substantial. For example, which bacterial phyla were reduced? What genes were significantly influenced? These need to be clearly described.

2. The body length and weight of experimental fish need to be added. Water quality indicators such as pH, ammonia nitrogen, nitrite and other data also need to be supplemented.

3. In the antibiotic exposure experiment, each group had only 1 glass aquarium and no biological duplication.

4. How many fish in each group were used in the antibiotic exposure experiment?

5. Sections 2.2 and 2.3 should be placed after section 2.4.

6. The significance analysis method is missing.

7. Section 3.5. The authors mentioned that several genes were up-regulated or down-regulated, and that data on the expression of these stress - and immune-related genes should be  supplemented.

8. Are transcriptome data validated? For example, using qPCR.

9. Vibrio parahaemolyticus decreased in the Ery group, but increased in the Amp. This phenomenon needs to be discussed.

Author Response

(The authors gave the same response as above.)

Round 2

Reviewer 1 Report

  • The present form of the manuscript is available for publication.

Reviewer 2 Report

The author made sufficient modifications according to the review comments, which can meet the requirements of publication.